# Secondary ARDS Following Acute Pancreatitis: Is Extracorporeal Membrane Oxygenation Feasible or Futile?

**DOI:** 10.3390/jcm10051000

**Published:** 2021-03-02

**Authors:** Mathias Schmandt, Tim R. Glowka, Stefan Kreyer, Thomas Muders, Stefan Muenster, Nils Ulrich Theuerkauf, Jörg C. Kalff, Christian Putensen, Jens-Christian Schewe, Stefan Felix Ehrentraut

**Affiliations:** 1Department of Anesthesiology and Intensive Care Medicine, University Hospital Bonn, 53127 Bonn, Germany; Mathias.Schmandt@ukbonn.de (M.S.); Stefan.Kreyer@ukbonn.de (S.K.); Thomas.Muders@ukbonn.de (T.M.); Stefan.Muenster@ukbonn.de (S.M.); Nils.Theuerkauf@ukbonn.de (N.U.T.); christian.putensen@ukbonn.de (C.P.); Jens-Christian.Schewe@ukbonn.de (J.-C.S.); 2Department of Surgery, University Hospital Bonn, 53127 Bonn, Germany; Tim.Glowka@ukbonn.de (T.R.G.); Joerg.Kalff@ukbonn.de (J.C.K.)

**Keywords:** extra corporeal life support (ECLS), extra corporeal membrane oxygenation (ECMO), pancreatitis, acute respiratory distress syndrome (ARDS)

## Abstract

Objective: To assess the feasibility of extracorporeal membrane oxygenation (ECMO) or life support (ECLS) as last resort life support therapy in patients with acute pancreatitis and subsequent secondary acute respiratory distress syndrome (ARDS). Methods: Retrospective analysis from January 2013, to April 2020, of ECMO patients with pancreatitis-induced ARDS at a German University Hospital. Demographics, hospital and ICU length of stay, duration of ECMO therapy, days on mechanical ventilation, fluid balance, need for decompressive laparotomy, amount of blood products, prognostic scores (CCI (Charlson Comorbidity Index), SOFA (Sequential Organ Failure Assessment), RESP(Respiratory ECMO Survival Prediction), SAVE (Survival after Veno-Arterial ECMO)), and the total known length of survival were assessed. Results: A total of *n* = 495 patients underwent ECMO. Eight patients with acute pancreatitis received ECLS (seven veno-venous, one veno-arterial). Five (71%) required decompressive laparotomy as salvage therapy due to abdominal hypertension. Two patients with acute pancreatitis (25%) survived to hospital discharge. The overall median length of survival was 22 days. Survivors required less fluid in the first 72 h of ECMO support and showed lower values for all prognostic scores. Conclusion: ECLS can be performed as a rescue therapy in patients with pancreatitis and secondary ARDS, but nevertheless mortality remains still high. Thus, this last-resort therapy may be best suited for patients with fewer pre-existing comorbidities and no other organ failure.

## 1. Introduction

Acute pancreatitis remains a life-threatening disease with an increasing incidence in Western countries [1,2]. It represents one of the most common diagnoses for hospital admission in patients with a gastrointestinal disorder [3]. The mortality rate ranges between 3–5% and has shown a downward contemporary trend [4,5]. In contrast, the incidence of acute pancreatitis for both outpatient contacts and hospital admissions has increased [6,7]. Frequent pathologies of acute pancreatitis are migrating gallstones, that cause transient obstruction of the pancreatic duct [8], and intense alcohol abuse [9,10]. Other causes include post-ERCP (endoscopic retrograde cholangiopancreatography) acute pancreatitis [11,12], and drug-induced acute pancreatitis [13]. About 20% of the patients with acute pancreatitis suffer from a severe course of the disease. [4,14]. Severity is defined as mild, moderate or severe, depending on transient (<48 h) or persistent (>48 h) organ failure or local or systemic complications [14]. Patients who develop persistent organ failure have significantly increased mortality with reported rates of 36–50% [15,16,17].

One of those organ failures is the development of secondary acute respiratory distress syndrome (ARDS). ARDS is an acute diffuse, inflammatory lung injury, leading to a substantial loss of aerated lung tissue and is clinically characterized by severe hypoxemia, bilateral opacities, and pulmonary edema that is not fully explained by cardiac failure or fluid overload [18]. In the Berlin definition of ARDS, severity is defined by the PaO_2_/FiO_2_ ratio and can range from mild, through moderate, to severe ARDS [18]. ARDS itself has a high mortality, with reported rates of up to 48% [18].

Intra-abdominal hypertension is a common complication in critically ill patients with acute pancreatitis that has the potential to progress into an abdominal compartment syndrome [19,20,21]. A decompressive laparotomy can be an option and should be indicated at the bedside by an interdisciplinary team approach [22]. Mostly, decompressive laparotomy is performed as rescue bedside laparotomy in the ICU because patients are not suitable for in-house transport to the operating room [22,23]. Most decompressive laparotomies are required due to abdominal conditions, with acute pancreatitis being one of the most common causes [24]. Decompressive laparotomy can be performed under extracorporeal life support (ECLS) as previously reported [22].

Current evidence that evaluates extracorporeal membrane oxygenation (ECMO) support in patients with acute pancreatitis and subsequent secondary ARDS remains poor [25,26,27]. All available published reports include a total of 16 patients. Most of these patients were treated before the availability of technically improved ECLS systems were available. ECLS systems that were available prior to the year 2008 suffered from major technical limitations. Here, bioincompatible membranes comprised the risk for major bleeding complications due to thrombocytosis and increased clotting within the ECMO membrane. This may have added to the high mortality rates that have been reported earlier. Today, further technical improvements of the ECMO membranes and circuits decrease the need for elevated anticoagulation levels, and therefore bleeding complications might be alleviated. Currently, there are only two published case reports available that have evaluated the treatment of acute pancreatitis and secondary ARDS with newer ECLS systems [28,29]. Further evaluation of ECLS for critically ill patients with acute pancreatitis and secondary ARDS is highly needed to support intensivists in the treatment of those patients.

Here, we report our single-center experiences for ECLS in patients with acute pancreatitis and subsequent secondary ARDS.

## 2. Experimental Section

We reviewed all ECMO (*n* = 495) case data from 1 January 2013, to 30 April 2020, from our institutional database. The analysis was approved by the local Ethics Committee and the need for individual informed consent was waived (Bonn Medical Faculty Ethics Committee #201/18).

Patients with a diagnosis of secondary ARDS due to acute pancreatitis were included (Figure 1). Data on demographics, comorbidities indicated by the Charlson Comorbidity Index (CCI) [30], the status of organ failure at the time of ECMO implantation indicated by Sequential Organ Failure Assessment score (SOFA) [31], and length of mechanical ventilation prior to ECMO initiation were collected. Parameters regarding the total length of stay in hospital and in intensive care were also recorded. To gain insights into the likelihood of survival from ECMO support therapy, the RESP [32] and SAVE [33] scores were used. We recorded total fluid balance within the first 72 h after initiation of ECMO support, overall ECMO duration, the success of ECMO weaning, general disease severity assessed by SAPS score [34] and nursing workload indicated by TISS score [35], total days of mechanical ventilation, and length of survival after hospital discharge. All values are presented as median ± interquartile Range (IQR). Data were analyzed using GraphPad Prism 8.0 (GraphPad Software, San Diego, CA, USA).

## 3. Results

Between 1 January 2013, and 30 April 2020, a total of 494 ECMO procedures were performed at our center. From those, eight ECMO patients with pancreatitis and pancreas-associated complications (i.e., secondary ARDS) were included in the analysis. Of those, seven were treated with veno-venous (VV), one was treated with veno-arterial (VA) ECMO. Two patients presented with necrotizing pancreatitis, all others initially with acute, non-necrotizing pancreatitis. One patient suffered from acute pancreatitis following partial pancreas resection. After developing secondary ARDS, the patient went into cardiac decompensation/low cardiac output syndrome alongside pancreatitis and was treated with VA-ECMO. Six patients were referred to us from other hospitals. Those patients were transported by our ECMO retrieval team and cannulated on-site before transport. Specifics regarding our ECMO mobile retrieval team have been reported before [36].

### 3.1. Demographics

Patients’ age ranged from 30 to 71 years (median 51, IQR 43–58). The median height, weight and Body Mass Index (BMI) values were 178 cm (175–180), 95 kg (82–110) and 28.4 kg/m^2^ (27.2–35.5), respectively. All patients were male.

### 3.2. Etiology of ARDS

Indication for ECMO: 5/8 patients suffered from hypercapnia (pCO_2_ > 75 mmHg) and were not responsive to ventilator therapy according to ARDS treatment recommendations and other conservative treatment approaches, including prone positioning (3/8 patients). Two patients were hypoxic with a PaO_2_ < 60 mmHg for >6 h. One patient, treated with VA-ECMO, suffered from acute myocardial infarction with low cardiac output syndrome following pancreatic surgery.

### 3.3. Etiology of Pancreatitis

Pancreatitis resulted from alcohol abuse (three patients), biliary (two patients), or an unknown cause (two patients). The patient-specific etiology is listed in Table 1.

### 3.4. Treatment Duration and Score-Based Outcome Prediction

Median in-hospital length of stay (LOS) and intensive care unit length of stay were 22 (10–54) and 22 (9–44) days, respectively. Two of the eight patients survived to hospital discharge, the remaining others could not be weaned from the ECMO circuit and finally deceased. The individual total time of survival is reported in Table 1. To predict in-house mortality for patients undergoing VV-ECMO, the RESP score was calculated for six of the seven VV-ECMO patients (one score was not determinable due to missing data). The median RESP score was −2, amounting to a predicted survival of 42.5%. The median SOFA score at time of ECMO initiation was 10 (8–11). Median SAPS and TISS scores at the initiation of ECLS were 49 (43–58) and 35 (30–41), respectively. Each individual’s scores are presented in Table 2. The median time on ECMO support was 10 days (1–23). Length of survival ranged from 1 to 518 days (22; 1–99). The median number of days on mechanical ventilation prior to ECMO was 3 (1–32), and days on mechanical ventilation in total amounted to 23 (8–44). Decompressive laparotomy due to abdominal compartment was performed in five patients according to the recommendations of the World Society of the Abdominal Compartment (WSACS). All patients were unsuitable for transfer to the operating theatre; therefore, decompressive laparotomy was performed at the bedside in the ICU. The two surviving patients did not receive decompressive laparotomy.

### 3.5. Blood Products and Total Fluid Balance

For seven patients, the amount of blood products (packed red blood cells, fresh frozen plasma/FFP, thrombocyte concentrates) administered were retrievable from the records. For Patient #1, this information was irretrievable. A median of 51 blood products (13–87) per patient was administered. More specifically, this resulted from packed red blood cells (46, 4–57), fresh frozen plasma (10, 4–18), and thrombocyte concentrates (3, 0–15) (the exact amount per patient can be found in Table 3). Additionally, PPSB (1000 IU, 0–4000), Factor VII (1000 IU, 0–3000), recombinant von Willebrand factor (vWF) (2400 IU, 0–7200), Factor XIII (5000 IU, 0–15,000), Fibrinogen (2 g, 0–4) and Antithrombin III (0–2000) were administered (detailed amounts per patient are presented in Table 4). Four out of the eight patients acquired von Willebrand Syndrome, i.e., a clinically relevant mismatch between vWF Activity and vWF-Antigen. Patients were treated with a combination of vWF/Factor VIII when the vWF-Activity:vWF-Antigen ratio was below 50% (severe). For mild cases (50% < vWF-Activity:vWF-Antigen < 75%), treatment was started when other signs of bleeding (e.g., from ECMO cannula sites, oropharyngeal mucosa, etc.) were present. Total fluid balance (i.e., sum of fluid intake and output) at Day 1 of ECMO treatment varied widely throughout the collective (498 mL; −7754–11,159 mL). This remained for the first three consecutive treatment days. The general treatment goal was the prevention of fluid overload or reduction in fluid volume to prevent peripheral edema, pulmonary edema, or other volume-associated side effects. The patients’ cumulative fluid balances and the daily fluid balances are reported in Table 5.

## 4. Discussion

Acute pancreatitis is a critical condition with high mortality [3]. Acute pancreatitis-associated mortality is further increased if complications such as secondary ARDS develop [15,17]. One of the hallmark characteristics of acute pancreatitis is a fluid overload, most often due to high volume resuscitation. This hypervolemia can lead to severe respiratory complications, sometimes mandating ECMO support. However, there is little evidence regarding the feasibility of ECMO support in adult patients who suffer from acute pancreatitis. Here, we report our experience from treating eight patients with acute pancreatitis and ECMO and try to highlight risk factors that are associated with poor outcome. Morbid obesity is a risk factor for acute pancreatitis [8,37] and significantly increases the severity in terms of morbidity, mortality, and medical expenses [4]. In our cohort, the median BMI was elevated above the norm, putting all patients into the overweight or adipose range. The surviving patients ranged at the lower end of the cohort (BMI 27 vs. median 28 kg/ms^2^), but differences are too small to draw any conclusions. Other factors negatively influencing the course of the disease are advanced age (≥60 years), severe coexisting conditions (≥2 on the Charlson Comorbidity Index score), and long-term alcohol abuse [8]. The importance of the CCI for ECMO patients has been previously reported by our group [22] and is mirrored here. The two surviving patients had no preexisting conditions (i.e., CCI 0), whereas all other patients had a CCI of ≥1. The sequential organ failure assessment score (SOFA) has been well established to correlate organ failure and prognosis [31,38]. The median SOFA score at the initiation of ECLS in our collective was 10, indicating a high level of combined organ failure and poor prognosis. The two surviving patients presented SOFA scores of 4 and 8, respectively. Hence, high SOFA scores should make the treating physicians critically re-evaluate the sensibility of starting a high-risk procedure such as ECMO.

All our patients received antibiotic treatment in accordance with the respective resistograms and the current guidelines. Antibiotic use in necrotizing pancreatitis was unable to lower mortality, the incidence of infected necrosis, or the need for operative treatment [39]. In acute pancreatitis, the literature reports a reduced rate of infectious complications with the use of Imipenem. However, the mortality and frequency of surgery were not affected [39].

Excessive bleeding and hemorrhage during ECMO treatment are frequent and increase mortality [40]. Bryner et al. and Kreyer et al. showed severe bleeding complications during ECMO support, in particular with prolonged ECLS runs. Here, we found no correlation between the occurrence of bleeding complications, the number of blood products used, or altered coagulation properties (e.g., extended PTT) through administered anticoagulants. The surviving patients showed acquired bleeding complications in the form of an acquired von Willebrand factor syndrome. However, the amount of blood product to control bleeding did not differ between survivors and non-survivors. Prolonged ECLS runs can now be performed without bleeding or coagulopathies being the limiting factor. This has been made possible by the use of newer, heparin-coated, biocompatible ECMO circuits, allowing for reduced usage of anticoagulants, thereby reducing the risk of fatal bleeding. However, the cohort size here is too small to directly draw conclusions.

Fluid balance, especially avoiding volume overload, appears to be of crucial importance in the critically ill [41]. However, under ECMO support, a positive fluid balance is often unavoidable to maintain sufficient ECMO blood flow for oxygenation. In the small cohort depicted here, patients with excessive fluid overload during the first 72 h died. The surviving patients achieved a negative fluid balance within the first 24 h.

Our observations are based on a male-only cohort. Whether this is due to a higher likelihood of males having an increased risk for complicated courses of acute pancreatitis remains a matter of scientific debate. Lankisch et al. state that “gender is no independent risk factor for the severity and outcome of acute pancreatitis” [42]. Other authors have proposed a higher risk for in-hospital death, gastrointestinal bleeding, and local complications among male patients with acute biliary pancreatitis [43]. Whether this holds true for other sub-forms of acute pancreatitis (e.g., alcohol-induced) remains unclear. Weitz et al., in a retrospective analysis of 391 cases of acute pancreatitis, stated that “Biliary, alcoholic and idiopathic acute pancreatitis should be treated as distinct entities” [44]. In their analysis, alcoholic etiology had a male predominance, with higher rates of necrosis [44]. Potentially, males, in combination with alcohol-induced acute pancreatitis, might have a higher rate of necrosis and thus might be at higher risk of decompressive laparotomy and non-favorable outcome. However, we believe our sample set to be too small to directly draw any conclusions from this focusing on gender.

Laparotomy does not further impair the already-reduced prognosis of patients undergoing ECLS [22], but it is known to significantly worsen outcomes of patients with acute pancreatitis compared to a step-up approach [45,46]. If laparotomy is inevitable, patients benefit from late necrosectomy (after 12 days compared to ≤3 days) [47]. The incidence of decompressive laparotomy in this sub-cohort of our ECMO cohort was high (5/8, ~62%) compared to the 5.2% (11/175) of all cases previously reported from our center [22]. However, the previous cohort did not include any acute pancreatitis patients. Partly, this might explain the higher incidence reported here. At our center, ECMO treatment for ARDS (159 patients in that study) without decompressive laparotomy was associated with a mortality rate of 45% [36]. This cohort comprised non-acute pancreatitis ARDS patients (mixed cohort of primary and secondary ARDS). Glowka et al. have reported that decompressive laparotomy itself is not associated with increased mortality when performed for abdominal compartment syndrome (ACS) alongside ECMO [22]. With a relatively low incidence of 5.2% (11/175), decompressive laparotomy was a rare occurrence. However, in patients receiving decompressive laparotomy alongside abdominal compartment syndrome under ECMO, 3/11 patients survived to hospital discharge, translating to a mortality of 73% [22]. It is very difficult to directly draw conclusions from this because the absolute occurrence of decompressive laparotomy reflects only a very small number of patients (*n* = 11) in the study by Glowka et al. and *n* = 5 in the current analysis. It is possible that decompressive laparotomy itself and not acute pancreatitis is the reason for increased mortality. Prospective studies in this regard are currently being performed and might help to answer this in the future.

Most strikingly, both surviving patients did not require/receive decompressive laparotomy. As Feddy and colleagues point out: “The lack of aggressive surgical intervention did not affect outcome”. In our case series, it is rather the opposite: the use of aggressive surgical intervention appeared to reflect negatively on the outcome. This highlights the need for restrictive fluid management, favoring vasopressors over volume administration. Furthermore, continuous or repeated measurements of intra-abdominal pressure (IAP) for early detection or increased IAP to prevent abdominal compartment syndrome should be performed, as previously recommended [22,26]. Decompressive laparotomy and subsequent surgery can be performed; however, the outcome in our experience is poor. Once ACS with the need for decompressive laparotomy is present, the continuation of ECMO appears to be futile. The use of ECMO at this point should be critically re-evaluated, because it is a very resource-intensive procedure, with the potential to divert resources from other critically ill patients, as highlighted in a case report by Rajdev et al. [48].

This study has some limitations: first and foremost, it was a single-center study with a very limited case number. Secondly, this cohort was very heterogeneous regarding etiology, and thus no conclusion can be directly drawn from it. However, considering the low incidence of ECLS with pancreatitis in our cohort (8/495; 1.6%), and the low number of published cases, it is unlikely, that any prospective studies with this focus will be launched. Therefore, we believe our findings to be of importance, especially because they comprise the largest collective treated with new generation ECLS systems and provide new data regarding bleeding complications, fluid intake, and potentially adverse effects of decompressive laparotomy alongside ECLS in patients suffering from secondary ARDS due to pancreatitis.

## 5. Conclusions

Bleeding complications are common but can be controlled, requiring high levels of blood products.

Prediction scores for survival (CCI, SOFA, RESP) should be utilized for decision making prior to ECMO initiation.

Avoiding excessive positive fluid balance within the first 72 h is favorable.

Bedside laparotomy for acute pancreatitis carried a 100% mortality and the procedure may not change the outcomes of ECMO for acute pancreatitis.

## Figures and Tables

**Figure 1 jcm-10-01000-f001:**
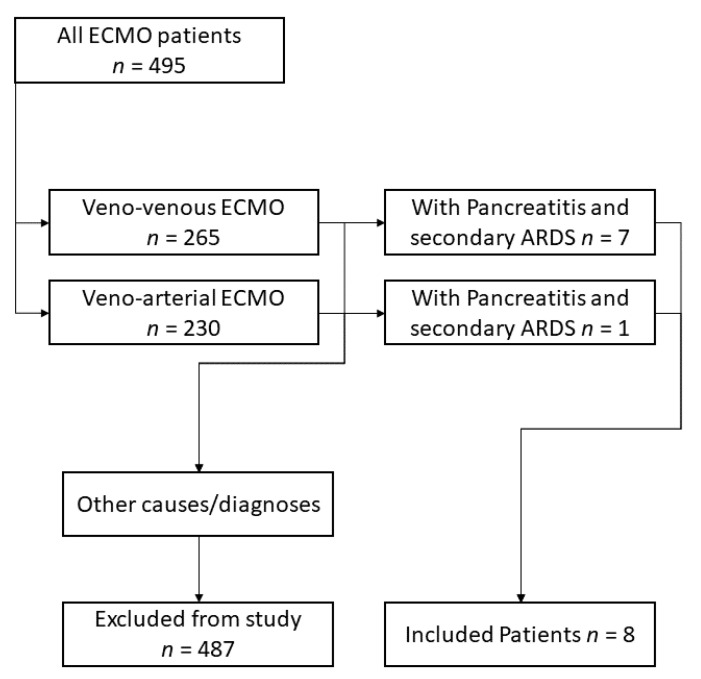
Inclusion process for the selected patient cohort. ECMO: Extra Corporeal Membrane Oxygenation, ARDS: Acute Respiratory Distress Syndrome.

**Table 1 jcm-10-01000-t001:** Patient characteristics.

#	Year Treated	Age at Hospital Admission, Years	in-Hospital LOS, Days	ICU LOS, Days	Height, cm	Weight, kg	BMI, kg/m^2^	Length of Survival, Days	Duration of ECMO, Days	Days on Mech. Ventilation Total	Days on Mech. Ventilation Prior to ECMO	Etiology	Laparotomy	Survived to Hospital Discharge
1	2014	51	20.0	19.7	175	75	24.49	20	19	20	0	Alcohol-induced	no	no
2	2019	51	69.0	69.0	180	94	29.01	29	29	70	41	Alcohol-induced	yes	no
3	2019	44	13.0	13.0	187	95	27.17	518	6	19	6	Alcohol-induced	no	yes
4	2019	30	44.0	44.0	175	150	48.98	1	1	44	43	Biliary	yes	no
5	2019	59	1.1	1.1	180	95	29.32	2	2	4	2	Unknown	yes	no
6	2020	71	9.4	8.0	170	79	27.34	1	0,7	2	1	Other, Post-pancreatic surgery	yes	no
7	2020	43	23.9	23.9	175	115	37.55	24	24	25	1	Biliary	yes	no
8	2020	56	56.8	42.7	180	90	27.78	112	14	43	4	Unknown, most likely biliary	no	yes

LOS, length of stay; ICU-LOS, intensive care unit length of stay; BMI, body mass index; ECMO: Extra Corporeal Membrane Oxygenation.

**Table 2 jcm-10-01000-t002:** Individual outcome prediction and disease severity scores.

#	CCI	SOFA	SAPS	TISS	RESP	SAVE
1	1	11	31	36	−3	
2	0	10	58	43	−5	
3	0	8	44	30	−1	
4	1	9	67	30	−4	
5	1	14	50	34	−1	
6	3	10	56	43		−8
7	6	8	42	35	0	
8	0	4	47	20	−2	

CCI, Charlson Comorbidity Index; SOFA, Sequential Organ Failure Assessment; SAPS, Simplified Acute Physiology Score; TISS, Therapeutic Intervention Scoring System; RESP, Respiratory ECMO Survival Prediction; SAVE, Survival after Veno-Arterial ECMO.

**Table 3 jcm-10-01000-t003:** Number of administered blood products.

#	Total Number of Blood Products	Red Blood Cell Concentrates	Fresh Frozen Plasma	Platelets
1				
2	177	89	65	23
3	6	6	0	0
4	87	57	15	15
5	13	4	4	5
6	14	4	10	0
7	70	49	18	3
8	51	46	4	1

**Table 4 jcm-10-01000-t004:** Anticoagulation, the occurrence of HIT, cumulative dose of coagulation factors.

#	Anticoagulation with	HIT	TXA	IU PCC, Cumulative	Recombinant VIIa	IU Factor VIII, Cumulative	rec. vWF, Cumulative	IU Factor XIII, Cumulative	Fibrinogen, Cumulative (g)	vWF-S	AT III, Cumulative (IU)
1											
2	Heparin (initially), than Argatroban	yes	yes	4500	400 kIU	2000	4800	7750	4	yes, mild	0
3	Heparin	no	no	0	0	1000	2400	0	0	yes, severe	0
4	Heparin (initially), than Argatroban	yes	yes	2000	0	0	0	5000	2	no	2000
5	Heparin	no	no	4000	0	0	0	2500	0	no	2000
6	Heparin	no	no	0	0	0	0	0	0	no	0
7	Heparin (initially), than Argatroban	yes	yes	1000	0	20,000	48,000	25,000	2	yes, severe	0
8	Heparin	no	yes	1000	0	3000	7200	15,000	4	yes, mild	0

TXA, tranexamic acid; rec, reconstituted; HIT, heparin-induced thrombocytemia; vWFs, acquired von Willebrand factor syndrome (mild: vWF-Activity/vWF-Antigen between 50–75%, severe: vWF-Activity/v-WF-Antigen <50%); PCC, prothrombin complex concentrate; IU, international units; AT, antithrombin.

**Table 5 jcm-10-01000-t005:** Fluid balance (mL) before ECMO.

#	Fluid Balance (mL) at Day 1 of ECMO	Day 2	Day 3	Cumulative (mL) for First 72 h
1	irretrievable			
2	−8205	−5776	−1438	−15,419
3	−3787	−1588	−292	−5667
4	7266	deceased		7266
5	2058	3743	deceased	5801
6	11,159	deceased	deceased	11,159
7	−7754	−1419	−2708	−11,881
8	498	−577	−1379	−1458

Fluid balance per day. All values are mL and reflect the daily fluid intake or output. Negative values reflect a net loss of fluids.

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
