# Peer review of "Secondary ARDS Following Acute Pancreatitis: Is Extracorporeal Membrane Oxygenation Feasible or Futile?"

_jcm, 2021, doi:10.3390/jcm10051000_

Round 1

Reviewer 1 Report

Author presented difficult complications of ECMO, which was acute pancreatitis.  It is difficult to manage on ECMO due to fluid balance issue and multi organ involvement due to acute pancreatitis.

I have read the author’s results differently.

Author concluded that “DLT alongside ECLS in patients with pancreatitis should only be considered as last resort.”  However, the author’s results showed zero survival among those who had undergone laparotomy according to Table 1.  From my personal experience, those who had bedside laparotomy for acute abdomen including pancreatitis during ECMO had mortality rate of 100%.  In my institution, we quit doing bedside laparotomy due to high mortality.  I do not agree laparotomy should reserve as last resort therapy based on your results and my personal experience.  The conclusion should be clearly written as “bedside laparotomy for acute pancreatitis carried 100 % mortality and the procedure may not change the outcomes of ECMO for acute pancreatitis”.

Please avoid abbreviation of “AP”  AP should be written as acute pancreatitis.  AP is not considered common abbreviation.

Please avoid abbreviation of “DLT”  DLT should be written as decompression laparotomy.  DLT is not considered common abbreviation.

Author Response

Reviewer 1: Author concluded that “DLT alongside ECLS in patients with pancreatitis should only be considered as last resort.”  However, the author’s results showed zero survival among those who had undergone laparotomy according to Table 1.  From my personal experience, those who had bedside laparotomy for acute abdomen including pancreatitis during ECMO had mortality rate of 100%.  In my institution, we quit doing bedside laparotomy due to high mortality.  I do not agree laparotomy should reserve as last resort therapy based on your results and my personal experience.  The conclusion should be clearly written as “bedside laparotomy for acute pancreatitis carried 100 % mortality and the procedure may not change the outcomes of ECMO for acute pancreatitis”.

Response: We thank the reviewer for his valuable time and effort to provde his review.

We adjusted the conclusion as per the reviewers request and agree, that it was, in its previous form, misleading. The reviewer stands correct, that we observed a 100% mortality for bedside laparotomy in this cohort. Our conclusion might have been biased from our experience with bedside laparotomy in a broader ECMO cohort (see Glowka et al., 2018). There, the results were not as unfavourable.

Reviewer 1:Please avoid abbreviation of “AP”  AP should be written as acute pancreatitis.  AP is not considered common abbreviation.

 Response: We omitted the abbreviation AP from the text and replaced it with “acute pancreatitis”. Changes are marked throughout the manuscript by the “track changes” mode

Reviewer 1: Please avoid abbreviation of “DLT”  DLT should be written as decompression laparotomy.  DLT is not considered common abbreviation.

Response: We omitted the abbreviation “DLT” from the text and replaced it with “decompressive laparotomy”. Changes are marked throughout the manuscript by the “track changes” mode

Reviewer 2 Report

Line 108- One patient, treated with VA-ECMO, suffered from acute myocardial 108 infarction with Low Cardiac Output Syndrome following pancreatic surgery.

can rephrase as "one patient who suffered from acute myocardial 108 infarction with Low Cardiac Output Syndrome following pancreatic surgery, was treated with VA -ECMO. 

Currently, it sounds like he developed cardiac injury on VA-ECMO.

Table-1 inhospital LOS, days ICU LOS, days. Why there are two kinds of numbers separated by a comma? 9,4 8,0. LOS should be one number i.e number of days unless I am missing something. 

Table-1- Days on vent/LOS etc are median or average? It is not consistent with the median numbers you have written in the following paragraph. 

Table4- expand all abbreviations, such as IE PPBS AT, etc.

Line 185-Solely for Imipenem a reduced rate of infectious 185 complications could be shown, while mortality and frequency of surgery was not affected.

Sounds odd, please consider rephrasing. 

Line 192- Here, we found no correlation between 'the' occurrence of bleeding complications or the 'number' of blood products and prolonged coagulation through administered anticoagulants. 

Fix grammar errors.

This sentence is unclear- you are saying found no correlation between which two entities? and what do you mean by 'prolonged coagulation through..??

Need thorough grammar check of the entire manuscript. Can use online tools like 'Grammarly'

193- The surviving patients showed acquired bleeding complications in the sense of an acquired von Willebrand Factor Syndrome.- consider in the "form' of an acquired ,,,

How did the patient achieve a 24 hr net balance of negative 7 to 8litres?

1. Rajdev K, Farr LA, Saeed MA, Hooten R, Baus J, Boer B. A Case of Extracorporeal Membrane Oxygenation as a Salvage Therapy for COVID-19-Associated Severe Acute Respiratory Distress Syndrome: Mounting Evidence. Journal of Investigative Medicine High Impact Case Reports. January 2020. doi:10.1177/2324709620957778   this report and literature review by Rajdev et al, discussed the PRESET and RESP scores, types of ECMO circuits, and highlights resource-intensive nature of ECMO. Consider citing this at the appropriate place where you have discussed ECMO circuits and scores. 

Author Response

Reviewer 2

Line 108- One patient, treated with VA-ECMO, suffered from acute myocardial 108 infarction with Low Cardiac Output Syndrome following pancreatic surgery.

can rephrase as "one patient who suffered from acute myocardial 108 infarction with Low Cardiac Output Syndrome following pancreatic surgery, was treated with VA -ECMO. 

Currently, it sounds like he developed cardiac injury on VA-ECMO.

 Response: Thank you for bringing this to our attention. We split the sentence into two separate sentences. By this, the order of events should be easier to follow (first pancreatic surgery, then secondary ARDS, then cardiac decompensation followed by VA-ECMO). The sentences are now: ll 104-108. One patient suffered from acute pancreatitis following partial pancreas resection. After developing secondary ARDS, the patient went into cardiac decompensation/low cardiac output syndrome alongside pancreatitis and was treated with V-A ECMO.

Reviewer 2: Table-1 inhospital LOS, days ICU LOS, days. Why there are two kinds of numbers separated by a comma? 9,4 8,0. LOS should be one number i.e number of days unless I am missing something. 

Response: We apologize for this mix-up. The numbers are the fractional days, not separate numbers. This is due to the different separator (English vs. German). Length of stay was calculated using the exact time of admission to time of discharge/death. Hence you see fractional days. 9,4 is 9.4 days, not two separate times, e.g. 9 or 4 days. Again, thank you for pointing this out. The comma separator was changed in table 1 accordingly.

Reviewer 2: Table-1- Days on vent/LOS etc are median or average? It is not consistent with the median numbers you have written in the following paragraph. 

Response: In table 1, the exact values per patient are presented. The text refers to the median number of the cohort.

Reviewer 2: Table4- expand all abbreviations, such as IE PPBS AT, etc.

Response: All abbreviations are now expanded within the table’s legend for quicker referencing. Thank you for this input.

Reviewer 2: Line 185-Solely for Imipenem a reduced rate of infectious 185 complications could be shown, while mortality and frequency of surgery was not affected.

Sounds odd, please consider rephrasing. 

Response: We reordered the paragraph. Hopefully this makes it easier to follow: All our patients received antibiotic treatment in accordance with the respective resistograms and the current guidelines. Antibiotic use in necrotizing pancreatitis was unable to lower mortality, the incidence of infected necrosis or the need for operative treatment [39]. In acute pancreatitis, the literature reports a reduced rate of infectious complications with the use of Imipenem. However, mortality and frequency of surgery was not affected [39].“

Reviewer 2: Line 192- Here, we found no correlation between 'the' occurrence of bleeding complications or the 'number' of blood products and prolonged coagulation through administered anticoagulants. 

Fix grammar errors.

Response: We corrected the errors and rephrased: “Here, we found no correlation between the occurrence of bleeding complications, the number of blood products used or altered coagulation properties (e.g. extended PTT) through administered anticoagulants

Reviewer 2: This sentence is unclear- you are saying found no correlation between which two entities? and what do you mean by 'prolonged coagulation through..??

Response: We changed and clarified, please refer to the previous comment.

Reviewer 2: Need thorough grammar check of the entire manuscript. Can use online tools like 'Grammarly'

Response: We corrected grammatical errors throughout the manuscript. All changes are highlighted in the text using the “track changes” option.

Reviewer 2: 193- The surviving patients showed acquired bleeding complications in the sense of an acquired von Willebrand Factor Syndrome.- consider in the "form' of an acquired ,,,

Response: We changed the sentence accordingly

Reviewer 2: How did the patient achieve a 24 hr net balance of negative 7 to 8litres?

Response: In patients with excessive fluid overload, this can often be seen. It is the combination of a high fluid output through multiple drainages (mostly through pleural catheters) and, once the endothelial barrier is reforming, the ability to run high ultrafiltrate rates using continuous renal replacement therapy. At that stage, we are sometimes able to achieve ultrafiltration rates of up to 600ml/h. As long as the patient tolerates this (i.e. no significant increases in lactate levels or excessive need for catecholamines), we continue to reduce fluid load.

Reviewer 2:

  1. Rajdev K, Farr LA, Saeed MA, Hooten R, Baus J, Boer B. A Case of Extracorporeal Membrane Oxygenation as a Salvage Therapy for COVID-19-Associated Severe Acute Respiratory Distress Syndrome: Mounting Evidence. Journal of Investigative Medicine High Impact Case Reports. January 2020. doi:10.1177/2324709620957778   this report and literature review by Rajdev et al, discussed the PRESET and RESP scores, types of ECMO circuits, and highlights resource-intensive nature of ECMO. Consider citing this at the appropriate place where you have discussed ECMO circuits and scores. 

Response: We considered this and appreciate the comment. While the case report nicely highlights the specific patients RESP/PRESET scores, we see no additional benefit from this in regard to the scores themselves. The aspect of resource intensiveness and the potential to divert/withdraw valuable resources away from other critically ill patients is, in my opinion, of far greater importance. We therefore included this into the discussion.

Round 2

Reviewer 2 Report

I appreciate the changes you have made. The manuscript looks much improved. No further suggestions. 

This manuscript is a resubmission of an earlier submission. The following is a list of the peer review reports and author responses from that submission.

Round 1

Reviewer 1 Report

This manuscript is prepared in sufficient way, but needs some revision to be clearer and understandable for even not specialist especially when dealing with abbreviations. In abstract for example, DLT and ECLS were not explained. Any abbreviation needs to be written in full name to be understood before use it. I wished if there was some additional statistical analysis as correlation to link some hospitalization services with the prognosis of the AP. Finding the population of patients as men only is interesting and it could be used to improved the quality of manuscript if there is other studies have considered the incidence of AP based on gender.

In the results, the authors found that ECLS-treated patients are having very low survival rate, but in conclusions they said it could be performed for patients who have ARDS as secondary to AP, which doesn't make sense; please remove first conclusion point.

Major revision:

Authors need to add some control data parallel to what they have listed in this manuscript, but it must be taken from non-AP ARDS patients who are utilizing ECMO (V-V and V-A ECLS) as life saver resort to compare and understand if the AP was the initial reason of poor survival rate. 

Author Response

Reviewer 1: This manuscript is prepared in sufficient way, but needs some revision to be clearer and understandable for even not specialist especially when dealing with abbreviations. In abstract for example, DLT and ECLS were not explained. Any abbreviation needs to be written in full name to be understood before use it.
Authors' reply: We thank the reviewer for her/his valuable insight and agree with the need for correct introduction of used abbreviations.
The abbreviation for decompressive laparotomy (DLT) was already introduced in line 19 of the original submission. ECLS has now been introduced in lines 14-15. Furthermore, we included a list of abbreviations at the end of the manuscript including all used abbreviations in alphabetical order and carefully rechecked the text.

Reviewer 1: I wished if there was some additional statistical analysis as correlation to link some hospitalization services with the prognosis of the AP. Finding the population of patients as men only is interesting and it could be used to improved the quality of manuscript if there is other studies have considered the incidence of AP based on gender.
Authors' reply: We agree, that questions regarding gender and “other services with regard to the prognosis of the AP” are plausible, however, our ICUs are primarily surgical ICU rather than medical ICUs and we are designated ARDS/ECMO center. Large number of patients are referred to us in our role as ECMO center. We are therefore not involved in any other “hospitalization services” and focus in this study on the treatment of secondary ARDS with ECMO. Hence we cannot comment on those. To acknowledge the reviewer’s question regarding gender influences as outcome predictor, we revised out Discussion section as follows (page 10, lines 214-225)
Of our analysed cohort (495 ECMO patients), all patients presenting with acute pancreatitis happen to be male. Thus, our observations are based on a male only cohort. Whether this is due to a higher likelihood of males having an increased risk for complicated courses of AP remains a matter of scientific debate. Lankisch et al. (Acute pancreatitis – Does gender matter?), state that “gender is no independent risk factor for the severity and outcome of acute pancreatitis”. Other authors have proposed a higher risk for in-hospital death, gastrointestinal bleeding and local complications among male patients with acute biliary pancreatitis (Shen et al.). Whether this holds true for other subforms of AP (e.g. alcohol induced) remains unclear. Weitz et al., in a retrospective analysis of 391 cases of acute pancreatits, state, that “Biliary, alcoholic and idiopathic acute pancreatitis should be treated as distinct entities” (Weitz et al.). In their analysis, alcoholic etiology had a male predominance with higher rates of necrosis (Weitz et al.). Potentially, males, in combination with alcohol induced AP might have a higher rate of necrosis and thus are at higher risk for DLT and non-favorable outcome. However, we believe our sample set to be too small to directly draw any conclusions from this focusing on gender.

Reviewer 1: In the results, the authors found that ECLS-treated patients are having very low survival rate, but in conclusions they said it could be performed for patients who have ARDS as secondary to AP, which doesn't make sense; please remove first conclusion point.
Authors' reply: We thank the reviewer for this insight and wholeheartedly agree. The mortality is exceedingly high, especially compared with non-AP ECMO recipients. The first conclusion point aims towards the technical feasibility. It is possible to perform ECMO support therapy and also control the therapy´s side effect like bleeding complications. Due to the high mortality, use of ECMO as bridging therapy should remain a last resort and potentially only under “compassionate use” certainly since ECMO itself is no cure. We rephrased the conclusion to better highlight this. In accordance with the paper´s title, ECMO is feasible (i.e. doable) but might still be futile. We hope that this sufficiently clarifies our statement. The rephrased conclusion now reads as follows: Page 11, lines 238-250

  • Bleeding complications are common but can be controlled, requiring high levels of blood products.
  • Prediction scores for survival (CCI, SOFA, RESP) should be utilized for decision making prior to ECMO initiation.
  • Avoiding excessive positive fluid balance within the first 72hours is favorable.
  • DLT alongside ECLS in patients with pancreatitis should only be considered as last resort.
  • ECLS can technically be performed in patients with ARDS secondary to pancreatitis. Hence, the use of ECMO in AP associated ARDS is feasible, but most likely futile. It should only be employed as “compassionate use” after all other options are exhausted.

Major revision:

Reviewer 1: Authors need to add some control data parallel to what they have listed in this manuscript, but it must be taken from non-AP ARDS patients who are utilizing ECMO (V-V and V-A ECLS) as life saver resort to compare and understand if the AP was the initial reason of poor survival rate. 

Authors' reply: We appreciate the reviewer’s opinion in regard to control data. In terms of survival rate, we conclude, that AP+DLT is the reason for poor survival. This stems from our previously reported data regarding ECMO transport and associated outcome. At our centre, ECMO treatment for ARDS (159 patients in that study) without DLT was associated with a mortality rate of 45% (Ehrentraut -  Interdisciplinary…). Said cohort comprised of non-AP ARDS patients (mixed cohort of primary and secondary ARDS). Glowka et al. previously reported, that decompressive laparotomy itself is not associated with an increased mortality when performed for abdominal compartment syndrome (ACS) alongside ECMO. With a relatively low incidence of 5,2% (11/175), DLT was a rare occurrence. However, in patients receiving DLT alongside Abdominal compartment syndrome under ECMO, 3/11 patients survived to hospital discharge, translating to a mortality of 73%. It is very difficult to directly draw a conclusion from this, since the absolute occurrence of DLT reflects only a very small number of patients ( n=11) in Glowka et al. and n=5 in the current analysis. It is possible, that DLT itself and not AP is the reason for increased mortality. Prospective studies in this regard are currently performed and might help answering this in the future.

We revised the Discussion and incorporated the aspects mentioned above on pages 10-11, lines 232-243.

Reviewer 2 Report

Dr. Schmandt from University hospital Bonn reported 8 cases of acute pancreatitis which utilized ECMO as a last resort. Only 2 cases discharged alive after ECMO management. It is difficult to define whether ECMO management for acute pancreatitis is feasible or not. This question has to discussed in the context of acute pancreatitis management. Thus these patients cohort has to be discussed within the whole cohort of acute pancreatitis (acute pancreatitis required hospital admission or required ICU care). Otherwise, this manuscript has to be submitted as simple case series. The main challenge of this field might be, how we address the risk of intraperitoneal bleeding which can be fatal in this patient population. Even we can manage hypoxia with veno-venous ECMO, the risk of intraperitoneal risk would be the seperate issues. Also, anticoagulation will increase the risk of bleeding. 

I believe this manuscript does not meet the expectation to be published as original article. However, as a case report, this manuscript can be published providing the evidence of ECMO management for acute pancreatitis even though the outcome is not very favorable.

Minor points:

  1. It would be better to provide how many patients admit to the hospital or ICU, how many required decompressive laparotomy. What is the mortality rate in each group to discuss the feasiblity of ECMO management.
  2. What is the reason, one patients required VA ECMO? If the patient need VA ECMO, the outcome might be even poorer. The early age (before 2008 as authors cited), VA ECMO was more common and the outcome was poorer with the needs of more strict anticoagulation.
  3. How you differentiate pure volume overload from ARDS. In this population, large amount of IV fluid is administered along the care. If patient with suboptimal heart function can cause pulmonary edema with large amount of IV fluid, which eliminate the diagnosis of ARDS.

Author Response

Reviewer 2: Dr. Schmandt from University hospital Bonn reported 8 cases of acute pancreatitis which utilized ECMO as a last resort. Only 2 cases discharged alive after ECMO management. It is difficult to define whether ECMO management for acute pancreatitis is feasible or not. This question has to discussed in the context of acute pancreatitis management. Thus these patients cohort has to be discussed within the whole cohort of acute pancreatitis (acute pancreatitis required hospital admission or required ICU care). Otherwise, this manuscript has to be submitted as simple case series.

Authors’ reply: We appreciate the reviewer´s opinion but beg to differ. In our opinion this is not a simple case series, meaning the aggregation of individual cases sharing a common feature, but was performed by analyzing 495 cases and screening those for acute pancreatitis in this cohort. From this large study cohort, only eight patients remained to be included in the final analysis. From this, we can only provide descriptive statistics, as mentioned in the Material&Methods section, and not perform comparative analysis of treatment groups. However, as others have already pointed out, given the rarity of the acute pancreatitis alongside ECMO, prospective studies might never happen. This is reflected by the scarcity of literature about this topic. Hence, we believe that our study should be regarded as an “original article”. Furthermore, the first reviewer stated no objections in regard to the submission of our work as “original article”. We leave the final decision to the editors esteemed discretion, but believe our work to fit this special issues scope to “present clinical and experimental scientific reports that improve our understanding of ARDS and provide information to improve “personalized” or “individualized” pharmacological and ventilator therapies in ARDS patients” (quoted from the call for papers).

Reviewer 2: The main challenge of this field might be, how we address the risk of intraperitoneal bleeding which can be fatal in this patient population. Even we can manage hypoxia with veno-venous ECMO, the risk of intraperitoneal risk would be the separate issues.

Authors’ reply: We agree with the reviewer that intraperitoneal bleeding can be a fatal complication of acute pancreatits. ECMO, in our experience, does not increase this risk, and we observed no fatal intraperitoneal bleeding in our study cohort.

Reviewer 2: Also, anticoagulation will increase the risk of bleeding. 

Authors’ reply: In respect to usage of anticoagulation (aPTT with a target of >45second) and its impact on bleeding, we agree with the reviewer. However, extended aPTT ranges were, in the past necessary due to bioincompatible ECMO membranes. Nowadays, advances in biotechnology allow usage of biocompatible, heparin-coated, circuits as standard treatment. Due to this, at our center, an aPTT of 35seconds is standard of care for vv-ECMOs. This reduces the amount of bleeding complications associated with ECMO and does not negatively impact on treatment success, as previously reported by Kreyer et al. This finding was just confirmed in a multi-centre study by Seeliger et al. (accepted for publication at Critical Care, “Comparison of anticoagulation strategies for veno-venous ECMO support in acute respiratory failure: a retrospective comparative cohort study" (CRIC-D-20-00878R2)). We incorporated the following sentence (page 10, lines 207-209) in the discussion: “This is now made possible by use of newer, heparin-coated, biocompatible ECMO circuits, allowing for reduced usage of anticoagulants, thereby reducing risk of fatal bleeding.”

Reviewer 2: I believe this manuscript does not meet the expectation to be published as original article. However, as a case report, this manuscript can be published providing the evidence of ECMO management for acute pancreatitis even though the outcome is not very favorable.

Please refer to our answer above regarding this.

Reviewer 2 Minor points:

  1. It would be better to provide how many patients admit to the hospital or ICU, how many required decompressive laparotomy. What is the mortality rate in each group to discuss the feasiblity of ECMO management.
    Authors’ reply: As pointed out in our reply to the first reviewer, the mortality rate and treatment of ECMO patients with and without decompressive laparotomy have been evaluated and reported in the analysis of Glowka et al. A prospective study is currently performed and the results are pending. Here we focus on secondary ARDS following acute pancreatitis under ECMO support since very limited data is available in this context. We therefore believe, that our focused contribution fits the aim of the special issue.

  1. What is the reason, one patients required VA ECMO? If the patient need VA ECMO, the outcome might be even poorer. The early age (before 2008 as authors cited), VA ECMO was more common and the outcome was poorer with the needs of more strict anticoagulation.
    Authors’ reply: We clarified, and rephrased, the section regarding the VA-ECMO patient’s etiology. VA-ECMO was performed to bridge low cardiac output syndrome, resulting from cardiac decompensation after secondary ARDS. The sentence now is as following: (page 3, lines 101-104) “One patient (treated with V-A ECMO) suffered from acute pancreatitis following partial pancreas resection and V-A ECMO was initiated due to secondary ARDS followed by cardiac decompensation/low cardiac output syndrome alongside pancreatitis

  1. How you differentiate pure volume overload from ARDS. In this population, large amount of IV fluid is administered along the care. If patient with suboptimal heart function can cause pulmonary edema with large amount of IV fluid, which eliminate the diagnosis of ARDS.
    Authors’ reply: We agree, that high levels of fluid administration during treatment of critical care patients in general (e.g. sepsis, acute pancreatitis, capillary leakage…) increases the risk of pulmonary edema. Hence, the etiology of pulmonary insufficiency leading to ECMO, might not be ARDS, but also compromised left ventricular function due to volume overload. However, all patients received transesophageal echocardiography to asses left ventricular function prior or during ECMO therapy. For the one patient requiring VA-ECMO, the time course was as following. First pancreatic surgery, then secondary ARDS, then left ventricular failure with the need for VA-ECMO. Knowing this and to the best of our knowledge we can rule out cardiac failure as reason for the initial pulmonary deterioration with near certainty.

The response to Reviewer 1 is attached in the separate word-file.

Round 2

Reviewer 1 Report

In my opinion, the manuscript now is ready to go!

Author Response

We thank the reviewer for her/his valuable time and effort and are pleased to have met his/her expectations. Thank you for now regarding the manuscript as publishable.

Mathias Schmandt and Stefan Ehrentraut

Reviewer 2 Report

The concerns I suggested is not well-addressed.

Author Response

Dear Reviewer 2,

we regret to hear, that our point-to-point reply did not meet your expectations. However, without further details, we feel unable to as how to respond. Unfortunately, you did not provide, what exactly did not meet your criticism.

Furthermore, it saddens us, that you state now, that the way the conclusions are supported by the results can be improved. In your first round of reviews, you indicated that this was fine and not needed any change.

Round 1:

  Yes Can be improved Must be improved Not applicable
Are the conclusions supported by the results? ( x) ( ) ( ) ( )

Round 2:

  Yes Can be improved Must be improved Not applicable
Are the conclusions supported by the results? ( ) (x) ( ) ( )

We'd like to express our thanks for your valuable time and effort you put into this review.

Mathias Schmand and Stefan Ehrentraut